# Numerical Simulation of Seepage and Stability of Tailing Dams: A Case Study in Ledong, China

**Jianxin Yang [1], Jun Hu [2] , Yuwei Wu [2,*] and Boyang Zhang [3]**

1 Hainan Geological Comprehensive Survey and Design Institute, Haikou 570206, China
2 School of Civil Engineering and Architecture, Hainan University, Haikou 570228, China
3 School of Material Science and Engineering, Zhengzhou University, Zhengzhou 450001, China
* Correspondence: wuyuwei@eis.hokudai.ac.jp

**Abstract:** Tailings dams are built to safely store tailings and to protect the natural environment from damage. However, tailings dam accidents occur frequently, endangering the safety of life and property, and causing pollution to the environment. Many tailings dam accidents are caused by seepage. As such, this study takes the No. II tailings dam of Ledong Baolun Gold Mine in Hainan Province as an example and builds a two-dimensional finite element model to simulate the seepage field. The effects of normal-water-level and high-water-level conditions on the total head, pressure head, and wetting line of the main and auxiliary dams were compared. The results show that higher water levels in both the main dam and the auxiliary dam lead to a higher pressure head at the top of the dam, lower pressure head at the bottom of the dam, higher total pressure head, and at the same time, a higher wetting line, and greater destabilization. In this study, the seepage deformation failure of the main dam and the auxiliary dam, in both cases, does not occur.

**Keywords:** tailings dam constructing; seepage; stability; numerical simulation; environmental control

## 1. Introduction

As global industrialization accelerates, the demand for ores is gradually increasing [1–3]. Large-scale mining and mineral processing generate large amounts of tailings as solid waste and sludge [4,5]. Tailings storage is the main method of solid waste disposal in China; by 2009, more than 12,000 tailings ponds had been built in China, and the number is increasing [6]. As artificial landfill slopes with high potential energy and large volume, tailings ponds are a major source of danger because they may form mudflows or large landslide disasters in the case of dam failure [7]. With social development and population migration, many tailings ponds downstream from where there were, at the time, no residents, villages, towns, factories, schools, railroads, and highways have now seen changes in situation. Additionally, once tailings ponds are disturbed, it not only threatens the safety of life and property of the surrounding and downstream residents, but there is also the fact that the leakage of tailings will cause serious pollution and ecological damage to the reservoir area and surrounding areas. Further, the damage to the environment is often long-term and difficult to recover from, and may even expand [8]. It is difficult to obtain the exact solution for the seepage field of tailings dams through theoretical study due to the complex geological conditions of the tailings dam and inaccurate boundary conditions. The most common methods of investigating the seepage of tailings dams are model tests and numerical simulations. The scale factor will, however, affect the model test. Additionally, the test results are heavily influenced by the test conditions, and model tests are expensive. As such, currently, the most common method to study the seepage field in tailings dams is by using numerical simulation. A study by Clark University, USA, showed that tailings dam accidents ranked 18th among the top 93 accidents and public hazards in the world [9,10]. There is an inherent tendency for seepage phenomena to occur at tailings

dams as a result of the combination of tailings and water [11]. A tailings dam can fail due to a variety of factors such as extreme events, foundation failure, overtopping, and poor operational practices [12–15]. Approximately 30–40% of tailings pond failures are caused by seepage and an increase in the phreatic surface [16,17]. As such, it is of great scientific importance to study seepage from tailings dams. As part of their investigation, McWhorter and Nelson examined one-dimensional seepage through tailings inside a reservoir and the effects of such seepage on groundwater [18]. Using a piezometric network, Stauffer and Obermeyer examined the pressure of pore water within tailings dams under different operational conditions [19]. Using SVFlux software, Rykaart et al. [20] modeled surface flux boundary conditions for water balance problems associated with tailings impoundments. To analyze seepage transients in tailings dams under the condition of extremely rapid water level decline, Ozer and Bromwell used limit equilibrium and finite element methods [21]. An approach for generalizing two-dimensional geological profiles was proposed by Zhao et al. [22], and the effect of generalized profiles on the seepage field was examined. As noted by Lu and Cui [23] and Zhang et al. [24], the two-dimensional model is not able to fully represent the complex and dynamic factors of a seepage field. Thus, their studies have suggested a new method of numerical modeling of the three-dimensional seepage field in tailings dams. This has been made possible due to the development of modern detection technology to detect damage to the horizontal impermeable layer of the tailings dam in real time. There have been some positive results achieved by the electrical leakage detection method in the analysis of construction damage and in the assurance of the quality of anti-seepage geomembranes [25]. Various factors contribute to the stability of a tailings dam, including building materials, tailings pond management, and tailings slurry composition [26]. As a result of other vulnerability factors, excessive high pore pressure can also accumulate during dam construction, reducing the effective stress and shear strength of tailings [27,28].

Based on the previous studies, this paper shows a detailed investigation of the tailings dam body that was carried out to identify the engineering and hydrogeological conditions and to also analyze the seepage stability of the tailings dam, using tailings reservoir No. II of the Baolun Gold Mine as the engineering background. This study is based on the finite element seepage analysis software GeoStudio 2007, SEEP/W model to fit the tailings dam infiltration line and to predict the infiltration line under special working conditions. The seepage analysis was performed using a saturated and unsaturated seepage instream model for two-dimensional seepage finite element analysis. Through drilling, a sampling test, in situ test, wave velocity test, and hydrogeological test, the engineering geological conditions of the tailings dam were identified. These geological conditions mainly included the stratigraphic lithology, tailings deposition pattern, and physical and mechanical properties of the dam. Further, based on the hydrogeological conditions of the tailings dam, the hydrogeological parameters of each geotechnical layer; the location; and the dynamic change characteristics of the tailings dam infiltration line were studied and—lastly—a seepage stability analysis of the tailings dam was performed.

## 2. Project Overview

Ledong Baolun Gold Mine, in Hainan Province, is located 235° from Ledong County (Baoyu Town), at a flat distance of 19 km. This mining site can be found at 108°59′00″ to 109°02′15″ east longitude and at 18°37′00″ to 18°41′15″ north latitude, as is shown in Figure 1a. The topography of the reservoir area is undulating, the original surface ditch in the reservoir area is developed, the overall topography is high in the southeast and low in the northwest, and the geomorphological unit of the site is hilly. The elevation of Haogangling to the southeast of the tailings dam is 500 m, and the elevation of Wanglou River valley to the west of the tailings dam is about 50–80 m, about 230 m from the tailings dam. The reservoir area is now covered with trees and shrubs. The steep canyons and seasonal drains are more developed in the site, and the original ground in the middle of the main dam and sub-dam comprises the seasonal drains. A schematic diagram of the main

and sub-dams of the stacking dam is shown in Figure 1b,c. The No. II tailings pond was completed and began operations in 2009. The original reservoir area of the tailings pond is about 0.47 km$^2$, and it is closed by the wet discharge method to 270 m elevations, with a total design dam height of 32 m and a corresponding total capacity of 426,800 m$^3$.

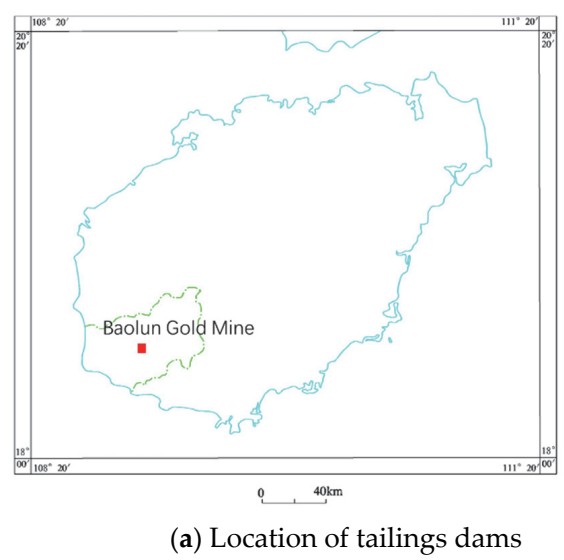

(**a**) Location of tailings dams

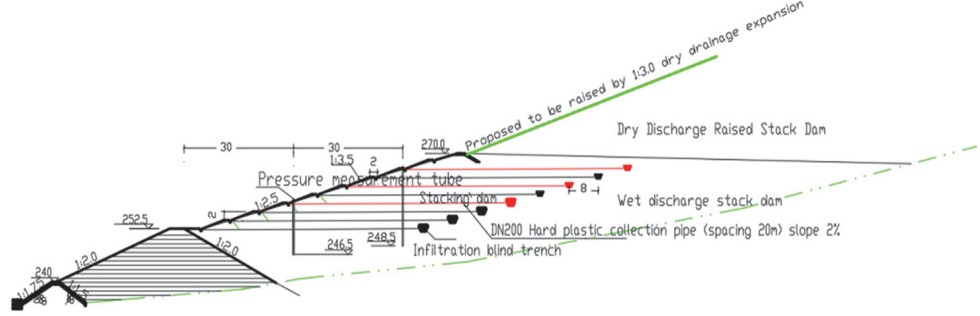

(**b**) Geological profile of the main dam

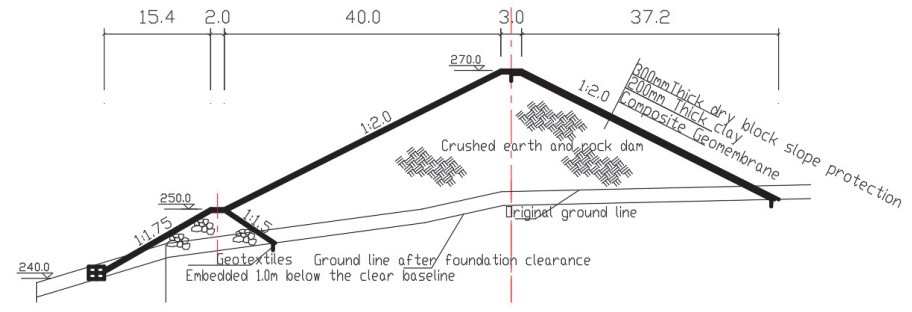

(**c**) Geological profile of the sub-dam

**Figure 1.** Diagram of the tailings pond of Baolun Gold Mine II.

## 3. Seepage Modeling Analysis of Tailings Dam

The bedrock beneath the reservoir area, and the dam site area, is mainly granite. According to the results determined from drilling, the joints and fissures of the medium-weathered granite near the surface are relatively developed (the cores are mostly in the form of cakes and fragments, with a small number of columns; half of the boreholes were drilled to the junction of the strong and medium weathering, and the whole hole was leaking). The results show that the granite joints and fractures in the dam site area are

mainly developed and can be categorized into two groups: north-east (dominant strike 40°, 65°, average dominant strike 53°) and north-west (dominant strike 295°); in addition, the density map of granite fractures in the dam site area shown in Figure 2 illustrate that the dip angle of bedrock fractures in the dam site area is generally greater than 45°.

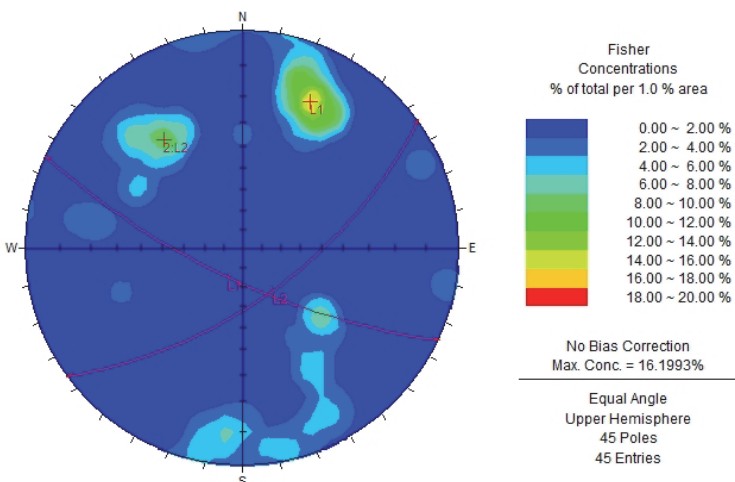

**Figure 2.** Isodensity map of granite joints and fissures in the dam site area.

### 3.1. Theoretical Representation of Numerical Models

As long as the groundwater changes extremely slowly, it can be regarded to be in a relatively steady state, that is, if the inflow and outflow of the cell are equal at any given time. An equation for steady seepage in two dimensions is given below.

$$\frac{\partial}{\partial x}\left(k_x \frac{\partial H}{\partial x}\right) + \frac{\partial}{\partial y}\left(k_x \frac{\partial H}{\partial x}\right) + Q = \frac{\partial \theta}{\partial t}$$

where $H$ is the total water head, $k_x$ is the permeability in the x-direction, $k_y$ is the permeability in the y-direction, $\theta$ is the volumetric water content, and $Q$ is the source.

The finite unit method uses the variational principle to find the generalized integral. The finite element form of the seepage control equations, in two dimensions, is derived using the Galliakin method with weighted residuals [29,30].

$$\tau \int_A \left([B]^T[C][B]\right) dA\{H\} + \tau \int_A \left(\lambda \langle N\rangle^T \langle N\rangle\right) dA\{H\}, t = q\tau \int_L \left(\langle N\rangle^T\right) dL$$

where $[B]$ is the gradient matrix; $\{H\}$ is the nodal head vector; $\langle N\rangle$ is the interpolation function vector; $q$ is the unit flow across the unit boundary; $\tau$ the is unit thickness; $A$ is the unit area; and $L$ is unit length.

The permeability coefficient function of unsaturated soils refers to the relationship between the permeability coefficient of water as a function of the particle size, density, suction, and structure of the soil. Permeability coefficient function models can be classified into three categories: empirical, macroscopic, and statistical [31]. Methods for measuring the permeability parameters of unsaturated soils are the steady-state test method, transient profile method, and soil–water characteristic curve method (specifically, a volumetric water content function estimation). In this study, the approximate function of the permeability coefficient in unsaturated soils, as proposed by Fredlund and Xing, 1994, is used [32], which is shown as follows,

$$k_w = k_s \frac{\sum_{i=j}^N \frac{\theta(e^y) - \theta(\psi)}{e^y i} \theta'(e^y i)}{\sum_{i=1}^N \frac{\theta(e^y) - \theta s}{e^y i} \theta'(e^y i)}$$

where $k_w$ is the permeability calculated under constant water content or matrix suction (m/s); $k_s$ is the saturated permeability (m/s); and $y$ is a dummy variable for the negative pore water pressure algorithm.

The soil–water characteristic curve represents the relationship between the matrix suction and the saturation of unsaturated soil. Based on the SWCC relationship proposed by Fredlund and Xing, the volumetric water content in unsaturated soil can be expressed as:

$$\theta_w = C_\psi \frac{\theta s}{\left\{ ln[e + (\frac{\psi}{a})^n]^m \right\}}$$

where $\theta_w$ is the volumetric water content, $C_\psi$ is the modified function, $\theta s$ is the saturated water content; and $a$, $m$, $n$ are the fitting parameters.

### 3.2. Basic Assumptions

To simplify the model and calculations, the following assumptions are made in this study.

(1) The permeability coefficient of the tailings soil does not differ much in horizontal and vertical directions, so it is regarded as an isotropic homogeneous material.

(2) The seepage problem, at various locations, is simplified to a laminar flow plane problem.

(3) Based on the infiltration line results from exploration observations, the initial dam water level is located below the top surface of the drainage body. As such, the initial dam is considered as the drainage prism of the stacked dam for calculation purposes.

(4) The permeability coefficient of the sub-dam of the stacked dam, which is located at the surface and has a small thickness, is similar to that of the tailings soil. As such, the influence of the tailings on the project is ignored.

(5) The sub-dam is composed of compacted earth and rock dam, its lower filler is gravelly clay in situ, its particle composition is not much different from that of the original gravelly clay, and its permeability coefficient function can be estimated by referring to the gravelly clay particle size distribution curve (permeability coefficient is taken as five times that of the original soil). The inner side of the sub-dam body is arranged with impermeable geotextile, which can be considered as a water barrier, etc. Bedrock is considered according to the water barrier boundary (i.e., the bedrock of the sub-dam section should be properly considered as the influence of the dominant direction of the tectonic fissure).

Simplified models use fewer computing resources while maintaining engineering accuracy.

### 3.3. Boundary Conditions and Material Parameters

Infiltration lines in the dam are high upstream, low downstream, high in the reservoir, and low in front of the dam; the annual seasonal variation of the water level at the reservoir end is less than 2.0 m. Due to the extraordinary variety of groundwater levels in extreme years, the trend of the tailings dam groundwater level continuing to raise the landfill, as well as certain safety reserves, the normal working condition is therefore elevated by 5 m. Accordingly, a water level of 270 is roughly consistent with the actual project, as determined by the tailings dam seepage analysis and static stability analysis of the high-infiltration-line conditions with the fixed head boundary conditions. Therefore, the project analysis and evaluation can be conducted safely and securely.

This study uses the volumetric water content function and saturation permeability coefficient to represent the soil permeability coefficient function. Based on the results of geotechnical tests such as saturated soil water content, volumetric compression coefficient, and particle size distribution curves, the volumetric water content function can be expressed. SEEP/W software is used to express the volumetric water content function based on the particle size data pair, while the Fredlund and Xing function is used to fit the correlation

between saturation and matrix suction. The main parameters and fitting parameters for the seepage analysis are shown in Table 1 as follows.

**Table 1.** Parameters and fitting parameters for the seepage analysis.

| Abbreviation/ Symbol | Parameter | Value Tailings Silts | Gravelly Clay Loam | Unit |
|---|---|---|---|---|
| $\theta_s$ | Volumetric heat capacity of the soil particles | 0.397 | 0.4535 | $m^3/m^3$ |
| $d_{10}$ | Thermal conductivity of the soil mixture | 0.005 | 0.003 | mm |
| $d_{60}$ | Material parameters accounting for the particle shape effect | 0.056 | 0.7 | mm |
| $W_L$ | liquid limit | 28.82 | 37.2 | % |
| $m_V$ | Volume compression coefficient of soil | 0.0001 | 0.000264 | $kPa^{-1}$ |
| $K_{sat}$ | Saturated water hydraulic conductivity | $2 \times 10^{-4}$ | $1 \times 10^{-4}$ | cm/s |
| $a$ | fitting parameters | 25 | 60 | kPa |
| $n$ | fitting parameters | 1.5 | 1.5 | 1 |
| $m$ | fitting parameters | 1 | 0.85 | 1 |

The grain size distribution curve and the soil water characteristic curve of the soil sample are shown in Figure 3.

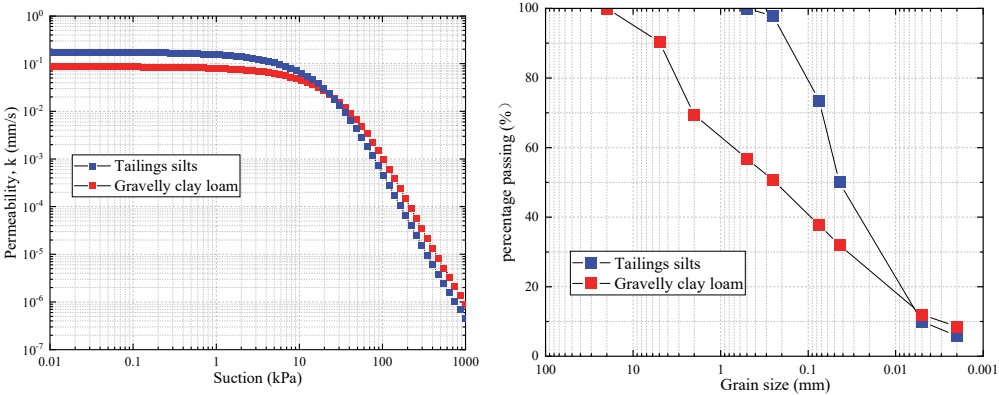

**Figure 3.** Grain size distribution curve and soil characteristic curve of the soil sample.

The tailings possess a high clay particle composition, in terms of the plasticity index of geotechnical statistics, as the average value of the plasticity index is 9.83 for dry discharge tailing soil and 9.54 for wet discharge tailing soil. In addition, with regard to the drilling cores, the cores can be intact, which also indicates their greater clayiness. Moreover, in terms of the particle distribution curve, the particle gradation curve is continuous, and more than 10% of its content is composed of clay particles that are less than 0.005 mm in size. At the same time, the average value of the Cu uneven coefficient of tailing powder soil is 11.3–14.2, but the particle size distribution curve of tailing powder soil is relatively steep when compared with the gravelly cohesive map, which indicates that the particle gradation of the tailing soil is average.

The particle gradation curve of the as-built gravelly clay loam is mostly smooth and continuous, and its fine-grained soil component (which is less than 0.075 mm) accounts for about 29.2% of its content. The particle size distribution curve shows that the content of less than 0.005 mm clay particles is more than 10%. The average value of the uneven coefficient cu is 330, which indicates that the gravelly clayey soil has good particle gradation in its original state. The cohesive soil in the dam filling of the main and sub-dams is gravelly

cohesive soil, and its particle size distribution is essentially the same as that of the original gravelly cohesive soil. However, the overall uniformity of the soil layer of the initial dam and the sub-dam is not as good as that of the original gravelly cohesive soil.

In the process of dam seepage stability analysis, it is necessary to set a safety criterion for seepage stability, which is usually expressed in terms of the allowable hydraulic ratio drop, the value of which is usually taken as a safety factor of 1.5–2.0 when based on the critical ratio drop. When infiltration has a bigger impact or hazard on the stability of the dam, it then takes the greater value. Further, the allowable hydraulic ratio drops without test data, and according to the empirical value of Table 2 the lower value is taken [33].

**Table 2.** Allowable Hydraulic Ratio Drop Table of Empirical Values.

| Allowed Hydraulic Ratio Drop | Permeability Deformation Type | | | | | |
| --- | --- | --- | --- | --- | --- | --- |
| | Fluid Type | | | Transition Type | Tube Surge Type | |
| | $C_u \leq 3$ | $3 < C_u \leq 5$ | $C_u \geq 5$ | | Cascading Continuity | Cascading Discontinuity |
| J | 0.25~0.35 | 0.35~0.50 | 0.50~0.80 | 0.25~0.40 | 0.15~0.25 | 0.10~0.20 |

In terms of the dam body tail powder soil and the dam base soil, gravelly clay soil particle size and its physical indicators can be obtained from Table 3, and as shown in this study, the different soil samples of the allowable hydraulic ratio drop.

**Table 3.** The allowable hydraulic ratio of dam body and dam foundation soil.

| Soil Sample | The Main Physical Indicators Used for Discrimination and Calculation | | | | | Type of Infiltration Deformation | Critical Hydraulic Ratio Drop | Allowable Hydraulic Ratio Drop | Location |
| --- | --- | --- | --- | --- | --- | --- | --- | --- | --- |
| | $C_u$ | Viscous Particle Content | $I_P$ | $G_S$ | $n$ | | | | |
| Tail powder soil | 11.3 | 11% | 9.83 | 2.66 | 0.397 | Fluid type | 1.00 | 0.50 | Dam body |
| Gravelly clayey soil | 330 | 13% | 13.63 | 2.67 | 0.4535 | Fluid type | 0.912 | 0.45 | Dam base soil |

The gradation curve of tail powder soil is noted as continuous, the content of clay particles exceeds 10%, and the plasticity index is 9.83, which means that its cohesiveness is good. It is judged to be non-tube-swelling soil, that is, the type of seepage damage of tail powder soil is flow soil type. In regard to gravelly clay soil for granite residual soil, of which its clay grain content is more than 10%, and powder grain clay grain content is between 35–40%. This is only because the clay soil contains a certain amount of not completely weathered siliceous soil, quartz and other hard particles that have been named above. The soil structure is close, iron oxide rich, and slightly cemented and with the formation of a reticulation structure, the soil is harder. The fine-grained soil and the coarse grains are cemented together; as such, its hydrological properties are essentially different from those of ordinary gravel soil. In summary, it is judged that non-tube surge soil, that is, the dam base gravelly clay soil seepage damage type, is mainly used for the flow of soil-type damage.

*3.4. Numerical Model*

The main dam is shown in Figure 4 with a 2-2' profile on the central axis as the seepage analysis profile. The seepage analysis of the main dam is carried out according to the geological exploration profile and the head boundary conditions. The profile meshes with an unstructured quadrilateral and triangular grid, and the approximate size of the global grid cell is 1.0 m. The geometry and mesh division of the model are shown in Figure 5. One

of the greatest advantages of unstructured meshing is that almost any shaped area can be divided into meshes.

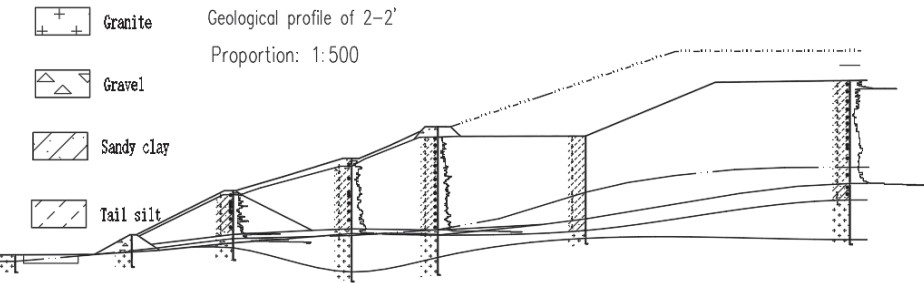

**Figure 4.** The 2-2 cross-section of the dam.

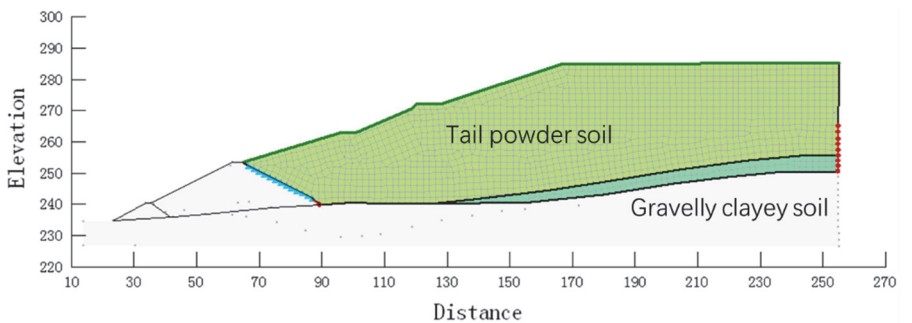

**Figure 5.** Main dam geometry model and meshing.

The sub-dam is a compacted earth and stone dam with an impermeable geotextile placed on the inner side of the dam. A 5-5' profile on the central axis as the seepage analysis profile of sub-dam, which is shown in Figure 6. A rock pile has been placed at the bottom of the dam to drain the seepage, which continues to the foot of the slope. There is a 2.5 m deep seepage pond at the bottom of the dam, a 2.5 m deep ditch to the south of the pond, and the slope itself is supported by a slurry stone retaining wall. The water depth in the ditch is about 0.1–0.2 m; further, there is a river 20 m west of the seepage pond at the bottom of the dam and the normal river level in the sub-dam section is 234 m and the maximum flood level is 236 m. The seepage analysis of the sub-dam was carried out according to the geological exploration profile and head boundary conditions. The profile has meshed with an unstructured quadrilateral and triangular grid, and the approximate size of the global grid cell is 1.0 m. The geometry and mesh division of the sub-dam are shown in Figure 7.

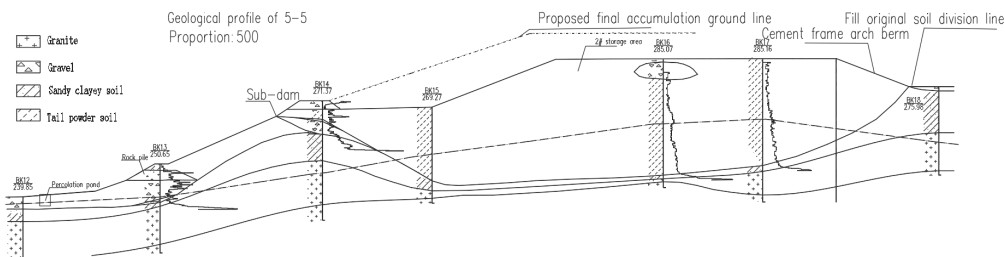

**Figure 6.** The 5-5 cross-section of the dam.

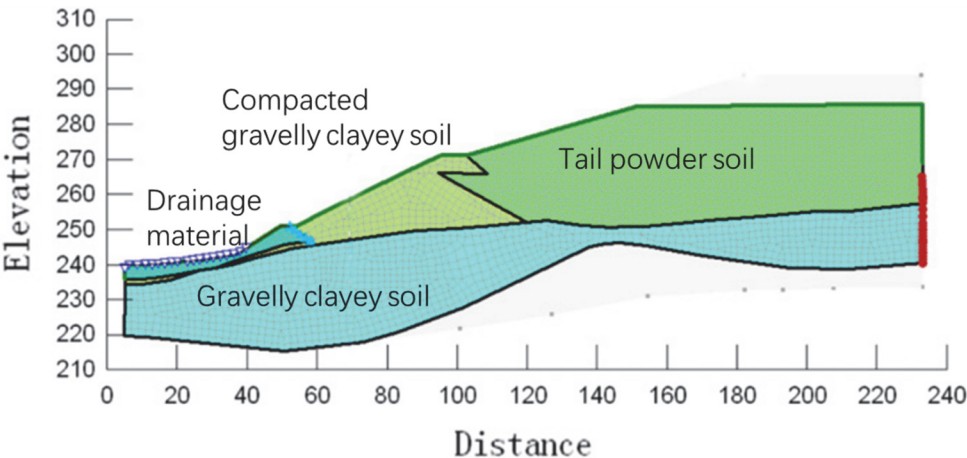

**Figure 7.** Sub-dam geometry model and meshing.

## 4. Results and Discussion

### 4.1. Main Dam

The above conditions were analyzed by using SEEP/W for 2D finite element seepage steady-state analysis. The total head distribution of the main dam under normal conditions, and with high-infiltration-line conditions, is shown in Figure 8.

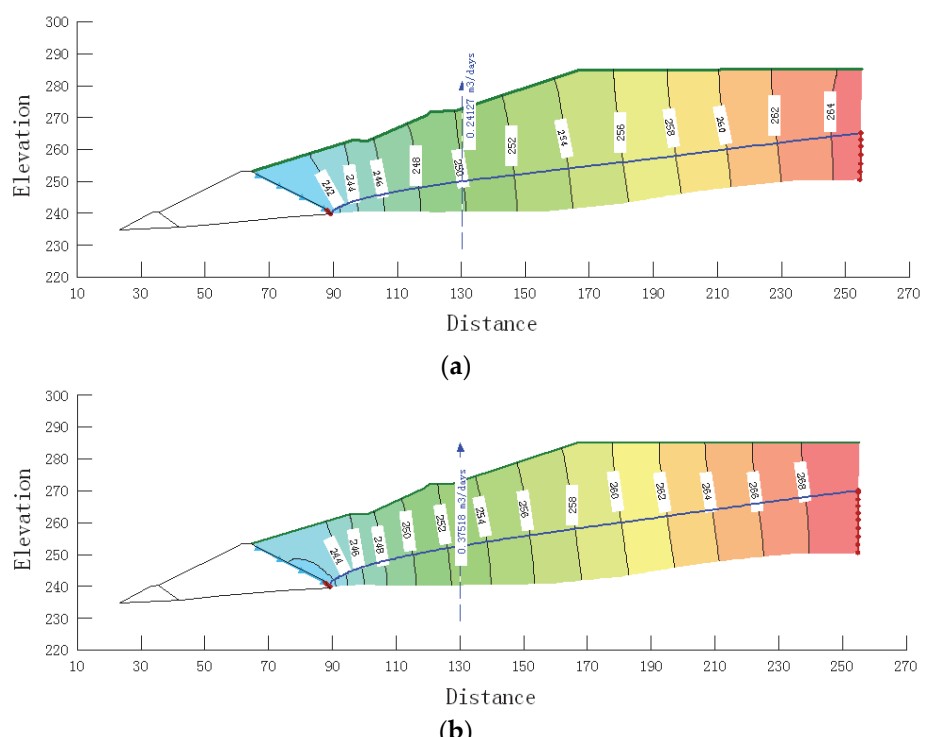

**Figure 8.** Total head distribution under different working conditions for the main dam. (**a**) Normal working condition total head distribution of the main dam (water level elevation: 265 m); (**b**) high-infiltration-line working condition total head distribution of the main dam (water level elevation: 270 m).

Figure 8 shows that the difference in the total head distribution between the normal working condition and the condition with a high infiltration line is not significant. The total head of the condition with a high infiltration line is slightly higher than the normal working condition. Figure 9 shows the pressure head distribution of the main dam under normal operating conditions and under conditions with a high infiltration line.

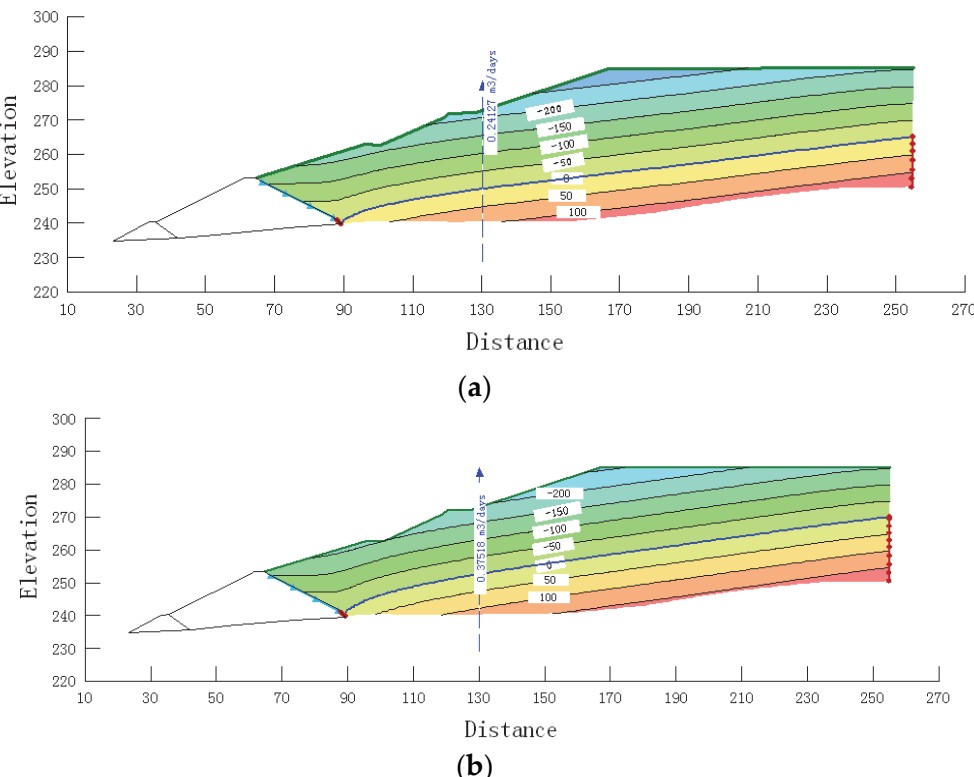

**Figure 9.** Pressure head distribution under different working conditions for the main dam. (**a**) Normal working condition pressure head distribution of the main dam (water level elevation: 265 m); (**b**) high-infiltration-line working condition pressure head distribution main dam (water level elevation: 270 m).

The isobaric line with a pressure head of 0 is considered the infiltration line under this working condition. As can be seen from Figure 9, the pressure head distribution under different operating conditions is recorded approximately. In the high-infiltration-line condition, the infiltration line of the main dam is higher, as such the water pressure at the top of the dam is lesser, and the water pressure at the bottom of the dam is greater when compared with the normal condition. Figure 10 shows the hydraulic gradient distribution of the main dam under normal and high-infiltration-line conditions.

As can be seen from Figure 10, the hydraulic gradient decreases more rapidly in terms of the high-infiltration-line condition when compared to the normal condition. Based on the above conclusions, a diagram of the infiltration line of the main dam under different working conditions is made, as shown in Figure 11.

As can be seen in Figure 11, the infiltration lines of normal and flood conditions were obtained via seepage analysis. According to the comparative analysis of the two conditions, it was found that when the water level at the end of the reservoir increased by 5 m, the water level in front of the dam changed by 0.5–1.0 m, i.e., the change in the groundwater level in front of the dam was smaller than that at the end of the reservoir. According to the seepage analysis, when combined with the tailings dam field survey mapping and dynamic observation results, the main dam in the two working conditions will not result in an escape of seepage either on the slope or the base of the dam infiltration line.

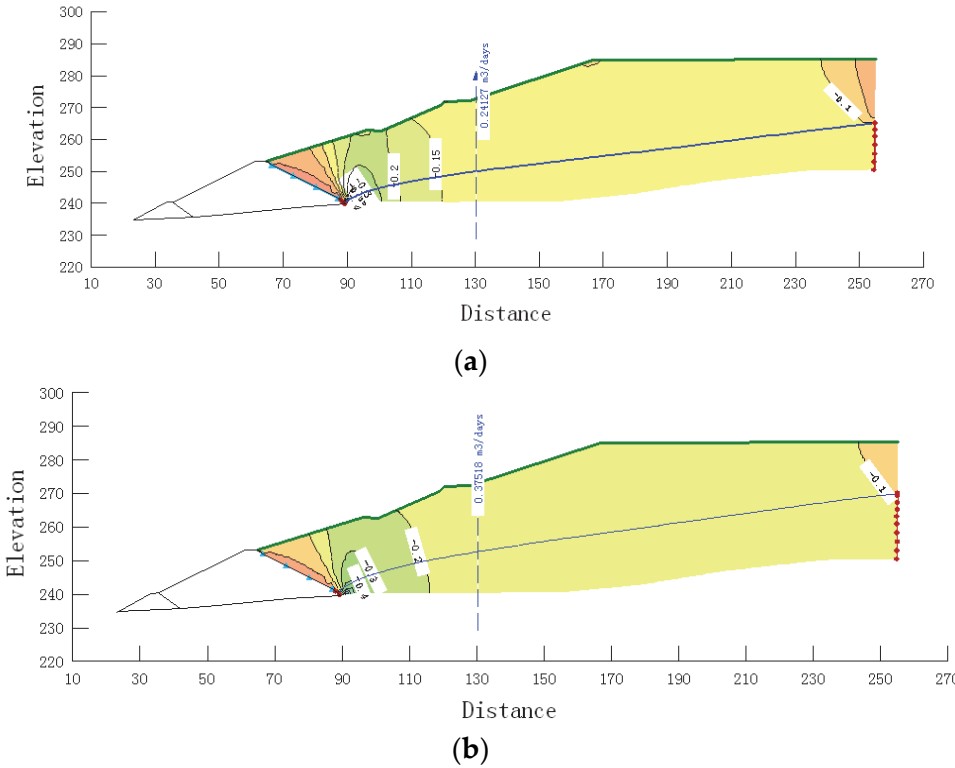

**Figure 10.** Hydraulic gradient distribution under different working conditions for the main dam. (**a**) Normal working condition hydraulic gradient distribution of the main dam (water level elevation: 265 m); (**b**) high-infiltration-line working condition hydraulic gradient distribution main dam (water level elevation: 270 m).

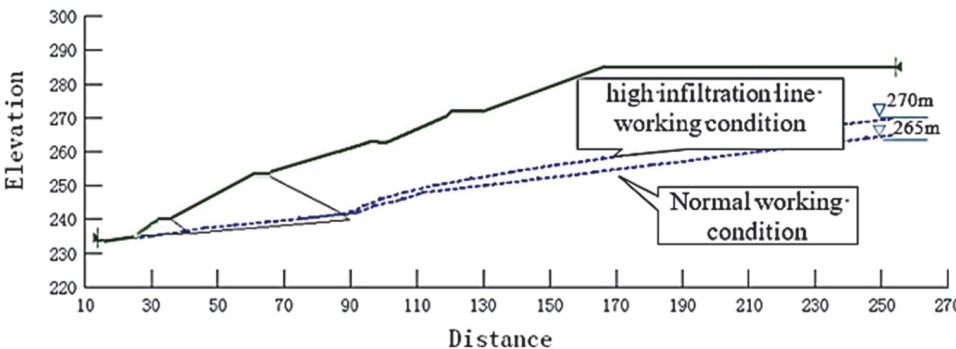

**Figure 11.** Main dam normal operating conditions, high infiltration line operating conditions infiltration line comparison chart.

### 4.2. Sub Dam

Similar to the main dam, the total head distribution, pressure head distribution, and hydraulic gradient distribution of the sub-dam under normal and high-infiltration-line conditions are discussed. Figure 12 shows the total head distribution of the sub-dam under normal working conditions and high-infiltration-line working conditions.

Similar to the main dam, it can be seen from Figure 12 that the total head distribution of the sub-dam has less variability under normal and high-infiltration-line conditions. The total head is slightly higher in the high-infiltration-line condition than in the normal condition. Figure 13 shows the distribution of the pressure head of the sub-dam under normal operating conditions and under conditions with a high infiltration line.

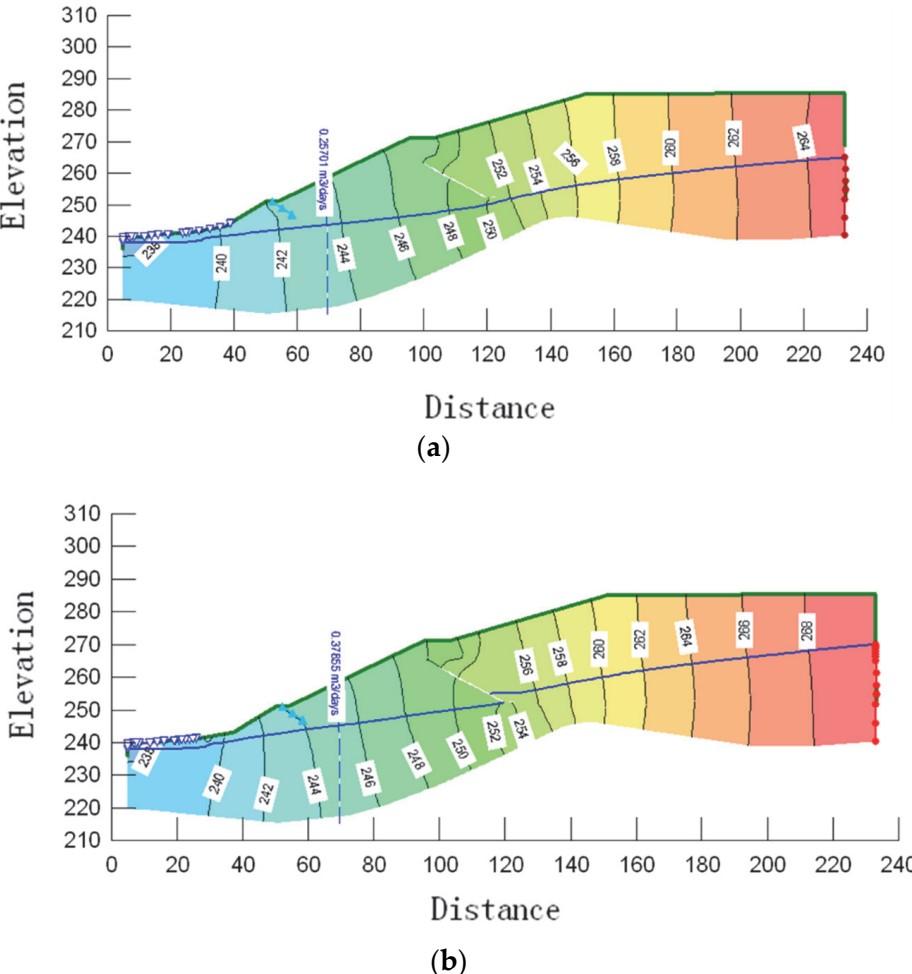

**Figure 12.** Total head distribution under different working conditions for sub-dam. (**a**) Normal working condition total head distribution of sub-dam (water level elevation: 265 m); (**b**) high-infiltration-line working condition total head distribution of sub-dam (water level elevation: 270 m).

The distribution pattern of the infiltration line and pressure head in Figure 13 is similar to that shown in Figure 9. The water pressure at the top of the dam is lesser, and the water pressure at the dam site is greater when the working condition with a high infiltration line is applied.

Figure 14 shows the schematic diagram that records the hydraulic gradient distribution of the secondary dam under different operating conditions. It can be seen that the hydraulic gradient is greater under high-infiltration-line working conditions than under normal working conditions. Additionally, a schematic diagram of the infiltration line under different working conditions is made for the sub-dam in order to compare the stability of the dam under different working conditions, as shown in Figure 15.

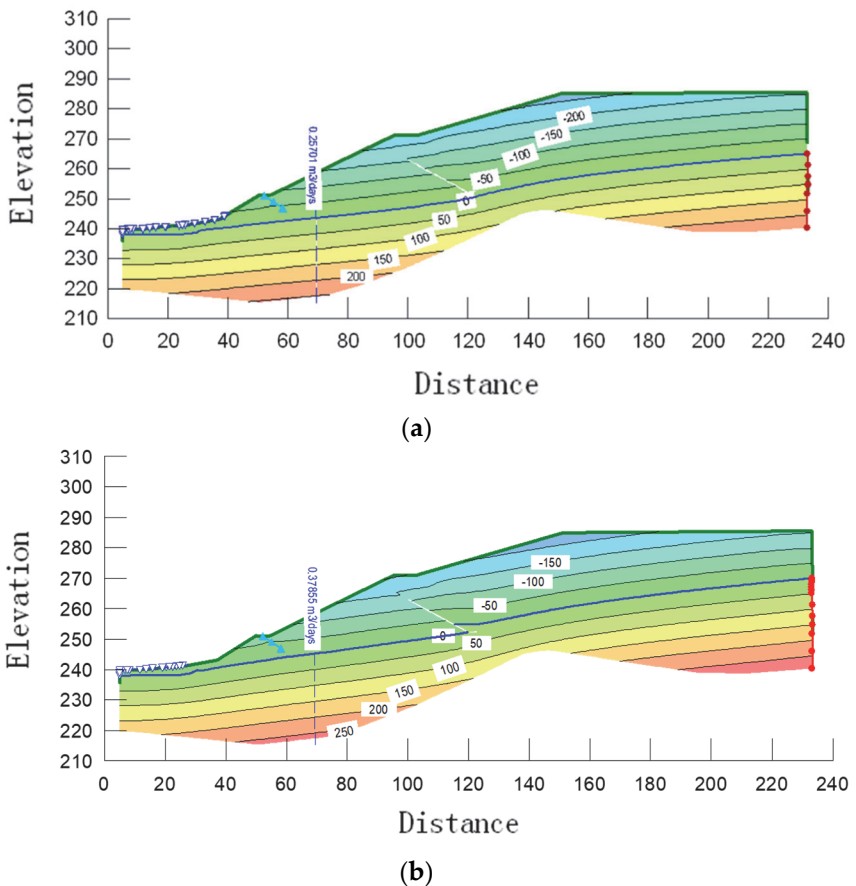

**Figure 13.** Pressure head distribution under different working conditions for sub-dam. (**a**) Normal working condition pressure head distribution of the sub-dam (water level elevation: 265 m); (**b**) high-infiltration-line working condition pressure head distribution of sub-dam (water level elevation: 270 m).

It can be seen from Figure 15 that the infiltration line of normal and special conditions was obtained through seepage analysis. According to the comparative analysis of the two conditions, when the water level at the end of the reservoir rises by 5 m, the water level in the middle of the sub-dam rises by 2.0–3.0 m. The water level at the bottom of the dam changes by 0.5–1.0 m due to the influence of the water barrier geotextile inside the dam., i.e., the groundwater level in front of the dam changes less than that at the end of the reservoir, which is consistent with the dynamic observation of the infiltration line under storm conditions. According to the seepage analysis obtained under normal and special conditions, the seepage volume per unit width is 0.257–0.378 $m^3$/d. The average calculated width of the main dam seepage is considered 20 m, and the calculated seepage volume obtained is 5.14–7.56 $m^3$/d.

The numerical simulation results show that under a normal water level and also a high water level, both the main dam and the sub-dam of the tailings dam are not damaged and deformed due to seepage. The results show that with the successful application of tailings dry-discharge technology during the construction of tailings dams can effectively ensure the stability of tailings dams. It also provides technical reference for the newly built tailings dam in Hainan Province and the extension and expansion of the same type of tailings dam.

Due to the geological and environmental conditions at the bottom of the sub-dam, a small portion of the tailings water seepage may enter the Wanglou River via the underground seepage, causing pollution to the river water. It is recommended that the foot of the dam be installed with an anti-seepage counter pressure, and, at the same time,

a waterlogging system should be arranged to reduce the infiltration line in order to reduce the tailings water seepage into the river.

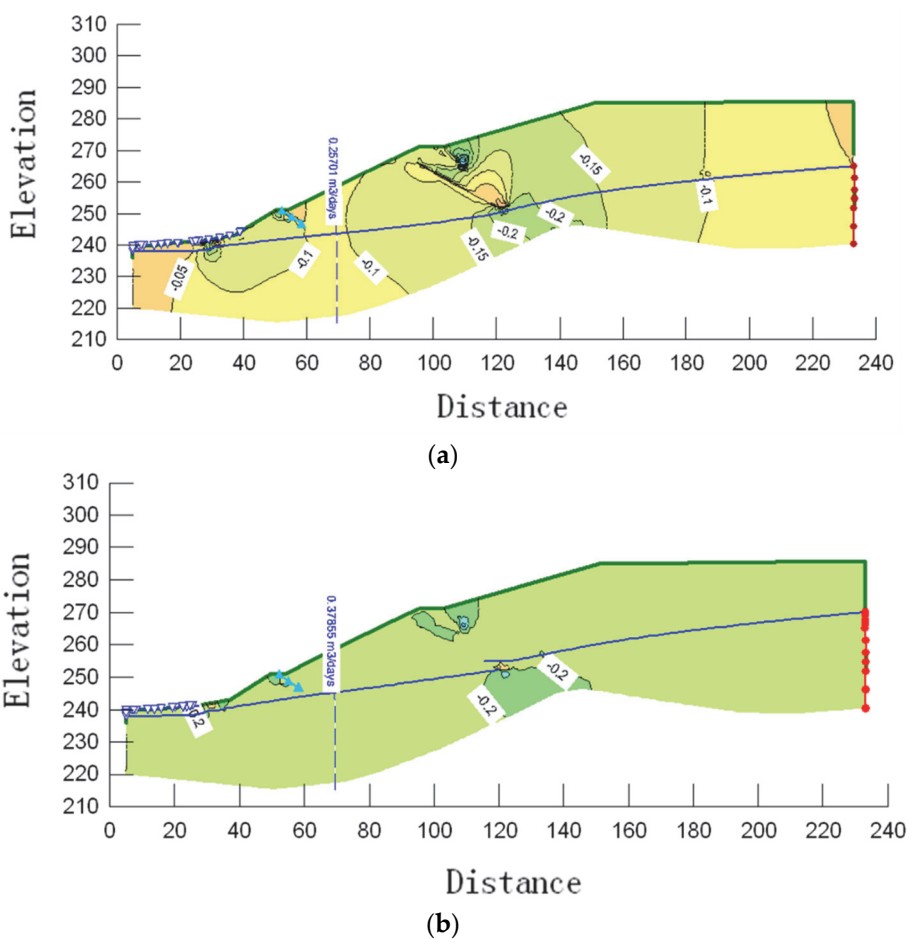

(**a**)

(**b**)

**Figure 14.** Hydraulic gradient distribution under different working conditions f for sub-dam. (**a**) Normal working condition hydraulic gradient distribution of sub-dam (water level elevation: 265 m); (**b**) sub-dam high-infiltration-line working condition hydraulic gradient distribution (water level elevation: 270 m).

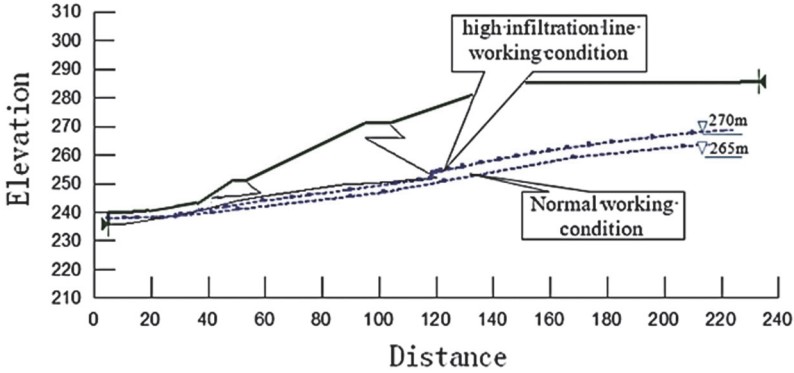

**Figure 15.** Sub-dam under normal operating conditions, high infiltration line operating conditions in an infiltration line comparison chart.

## 5. Conclusions

Seepage and stability calculations are important for the design and construction of tailings dams. In this study, an engineering overview of the tailings dam of the Ledong

Baolun Gold Mine in Hainan Province is provided. A finite element model was developed based on the engineering data of the Lixi tailings dam. The seepage and stability were calculated, the infiltration lines for normal and special conditions were obtained. Further, the critical hydraulic specific drop and allowable hydraulic specific drop of the tailing powder soil (tailings) and gravelly clay soil (dam base) were calculated. The main conclusions are summarized as follows.

1. The tailings pond infiltration line is low and insensitive to heavy rainfall; the infiltration line variation is mainly controlled by seasonal rainfall and is generally smooth, albeit with a slightly larger variation inside the pond and a relatively smaller variation at the bottom of the dam.

2. In the computed sections of the main dam, the coefficient of safety for slip resistance under various operating conditions is greater than the minimum safety coefficient allowed by the code, which means that the main dam section has the basic conditions for raising and expanding the capacity.

3. According to the two-dimensional finite element steady-state permeability analysis of the main and sub-dams, seepage will not occur under the current and flood conditions. Moreover, the hydraulic gradient obtained from the seepage analysis is less than the hydraulic gradient that is allowed for the flow-type damage of the relevant soil layer; further, no seepage damage such as flow damage will occur in the dam body and dam foundation.

With the rapid development of machine learning techniques in recent years, tailings dam erosion research based on machine learning has been further developed as well. Yang's study proposed a CNN–LSTM network to predict the risk of tailings dams, which can study the trend of the saturation line and issue early warnings [34]. Li [35] proposed a region growth segmentation algorithm based on multiple sub pixels combined with point cloud coordinates for accurate identification of deformed areas of tailings dams. Li [36] developed a deep learning-based target detection method for identifying the location of tailings ponds and automatically acquiring their geographical distribution from high-resolution satellite images. These research results allow for the timely and accurate mapping and monitoring of tailings dams, which is crucial for decision makers to improve management methods and prevent disasters caused by tailings dam failures. Consequently, we plan to utilize image recognition in future studies in order to calculate and recognize tailings dam instability.

**Author Contributions:** Conceptualization, J.Y. and J.H.; methodology, J.Y.; software, J.Y.; validation, J.Y., J.H. and B.Z.; formal analysis, J.Y.; investigation, J.Y.; resources, J.Y.; data curation, J.Y.; writing—original draft preparation, Y.W.; writing—review and editing, Y.W.; visualization, J.Y.; supervision, J.Y.; project administration, J.Y.; and funding acquisition, J.Y. All authors have read and agreed to the published version of the manuscript.

**Funding:** This research was funded by the Innovative Research Team Project of Natural Science Foundation of Hainan Province, P. R. China (522CXTD511), the High Technology Direction Project of the Key Research & Development Science and Technology of Hainan Province, P. R. China (ZDYF2021GXJS020), the High-level talent project of Hainan basic and applied basic research plan (2019RC148, 2019RC351), the Hainan Natural Science Foundation, (519QN333), and the characteristic innovation (Natural Science) projects of scientific research platforms and scientific research projects of Guangdong Universities in 2021(2021KTSCX139). The authors also acknowledge support from the China Scholarship Council Project (201907565040).

**Institutional Review Board Statement:** Not applicable.

**Informed Consent Statement:** Not applicable.

**Data Availability Statement:** Not applicable.

**Acknowledgments:** The assistance by Jia Lin from Hainan Investigation Institute of Hydrogeology and Engineering Geology is greatly appreciated.

**Conflicts of Interest:** The authors declare no conflict of interest.

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
