# Peer review of "Numerical Simulation of Seepage and Stability of Tailing Dams: A Case Study in Ledong, China"

_sustainability, doi:10.3390/su141912393_

Round 1
Reviewer 1 Report
The specific modifications are as follows.
(1)The paper discusses the seepage of the tailing dam but does not discuss the stability of tailing dams in detail, so it The research on the literature is not rich enough.
(2)It is suggested to add some literature on the modification of tailings by using interdisciplinary technology.is suggested to revise the title of the article.
(3)There are some spelling errors in the article, such as the title of Figure 3 in line 184.
(4)The format of Tables 2 and 3 is not standardized.
(5) More discussion is needed. In the current format, the provided discussion doesn't show the driving phenomena.
Reviewer 2 Report
The authors present a general interesting study on stability and prevention of tailings No. 2 of Baolun 2 Gold Mine in Ledong of China. The reviewer has a list of suggestions and comments for the authors to consider in a minor revision before the paper could be accepted.
1. Line 84-90, in Fig. 1a, the color of the mark “Baolun Gold Mine” is not so obvious and the watermark “2022 Google” in the picture could be erased, and there is a symbol “????” should be clarified in Fig. 1b.
2. Line 157-158, whose and why permeability coefficient is taken as 5 times of the original soil? Please clarify that.
3. Line 179, the title of table 1 is missing. As requested by the journal, the title should be included by the author.
4. Some of the expressions in the paper are not standardized. The symbols of "greater than" and "less than" need to be expressed in words, and the expressions used in other parts of the article need to be corrected accordingly.
5. It should be noted, however, that the English of the abstract is not excellent and, in addition, sometimes it fails to accurately reflect the content of the paper. It is stated, for instance, that "the location and dynamic change characteristics of the tailings dam infiltration line were obtained." However, the paper does not address the dynamic changes characteristics of tailings dam infiltration lines, as it only shows a single example of tailing dam infiltration lines. In addition, "the seepage stability analysis of the tailings dam indicates that the use of tailings dry drainage technology in tailings dam construction has effectively ensured the tailings dam's stability. " This paper does not provide a detailed explanation of the technology "tailings dry drainage technology".
6. Line 379, is the sentence “Lidong Baolun Gold Mine II in Lihai Province” referring to the “Ledong Baolun Gold Mine in Hainan Province” in project overview?
7. In the manuscript, there are many format and grammar errors like “the tailings” should be capitalized in line 386, a full stop “.” is needed in line 376 and the unknown reference “[58]” in line 125, and so on, it is suggested to read through and check carefully.

Reviewer 3 Report
The work submitted to the MDPI Sustainability entitled "tudy on stability and prevention of tailings No. 2 of Baolun Gold Mine in Ledong, Hainan" is reviewed.
The proposed study investigates the case study of the No. 2 tailings dam at the Baolun gold mine in Ledong, Hainan. A two-dimensional finite element model is constructed where the seepage field of the project was simulated, and the location and dynamic change characteristics of the tailings dam infiltration line were obtained. Finally, the seepage stability analysis of the tailings dam was performed.
The reviewer believes this research paper could be interesting to the hydrological, geotechnical and geological research community and those interested in sustainability.
In general, the paper is well structured, and the data is well analyzed; however, it requires further discussion at a few points. However, I am suggesting the manuscript be accepted for publication from the Archives of Civil and Mechanical Engineering if the authors are willing to perform major improvements/corrections on the submitted work.
Here are the major improvements/corrections I suggest authors to review:
- Article need rigorous English proof reading.
- Title of the paper should be re considered. Something like “Numerical simulation of seepage and stability of Tailing Dams: Case Study in ……
- Some numerical findings should be included in the abstract.
- Pg 2, L 60 – Please mention the software version used in this study.
- Pg 2, L 70 – Coordinates of the study area should be given.
- Figure 1 needs to be updated to make it clear. Please delete unnecessary lines and text.
- Pg 2, L 92-94 – It seems that some text is left over from the template. Please delete the paragraph.
- In section 3.2 author list some assumptions taken while performing the calculations. Authors should state how those assumptions will influence the outcomes.
- Pg 5, L 172-178 authors stated that the geotechnical parameters such as volumetric water content was predicted by using the software. How can the authors be sure that correlated parameters are accurate enough.
- -Pg 5, L 180 table does not have a caption. Also mention the standards followed to obtain the properties listed in Table .
- Please check the table and figure captions they are not all written with same font. Also some figures have missing units.
- Please check the referencing style both in text and at reference section.
- It is advice to add a figure of experimental sample during preparation, testing arrangement and failed sample.
- Before listing the conclusions, a paragraph required that reflects authors personal reflections on results.
- Please add some recommendations for future studies and limmitations of this study to end of conclusions section.

Round 2
Reviewer 3 Report
Authors have successfully tackled all the comments given by the reviewers. I am suggesting the manuscript to be accepted for publication from the Sustainability.